# PEPscan: A Broad Spectrum Approach for the Characterization of Protein-Binder Interactions?

**DOI:** 10.3390/biom12020178

**Published:** 2022-01-21

**Authors:** Angelita Rebollo, Louise Fliedel, Pierre Tuffery

**Affiliations:** 1Université de Paris, INSERM U 1267, CNRS U 8258, UTCBS, F-75006 Paris, France; angelita.rebollo@parisdescartes.fr (A.R.); louise.fliedel@parisdescartes.fr (L.F.); 2Unité de Biologie Fonctionnelle et Adaptative, Université de Paris, CNRS U 8251, INSERM U 1133, F-75013 Paris, France

**Keywords:** PEPscan, protein-protein interaction, protein-peptide interactions, protein-polysaccharide interactions

## Abstract

In a previous study, we have shown that PEPscan can provide a cheap and rapid means to identify candidate interfering peptides (IPs), i.e., peptides able to disrupt a target protein-protein interaction. PEPscan was shown to be effective in identifying a limited number of candidate IPs specific to the target interaction. Here, we investigate the results of 14 new PEPscan experiments for protein complexes of known 3D structures. We show that for almost all complexes, PEPscan is able to identify candidate IPs that are located at the protein-protein interface. The information it provides about the binding site seems, however, too ambiguous to be exploited in a simple manner to assist the modeling of protein complexes. Moreover, these candidates are associated with false positives. For these, we suggest they could correspond to non-specific binders, which leaves room for further optimization of the PEPscan protocol. Another unexpected advance comes from the observation of the applicability of PEPscan for polysaccharides and labeled peptides, suggesting that PEPscan could become a large spectrum approach to investigate protein-binder interactions, the binder not necessarily being a protein.

## 1. Introduction

Peptide arrays have a wide range of applications in basic and applied research. The arrays consist of hundreds of different peptide sequences immobilized on a solid support, which are analyzed in terms of spots associated with a signal. Their usage differs depending on the choice of the peptide synthesis method, the solid support, the immobilization method, the size of the peptides, the density of peptides on the solid support, and the detection method. The most widely used technique to generate peptide arrays is the SPOT synthesis. This technique is based on solid-phase Fmoc chemistry to synthesize peptides on a membrane support, which is usually nitrocellulose [1,2,3]. The majority of the peptide arrays are analyzed using labeling-dependent assays, and the most common methods are chemiluminescence, colorimetry, and fluorescence. Peptide arrays offer many possibilities to analyze different signaling pathways between normal and pathological conditions [4]. They have been used for antibody epitope mapping [5], the identification of the binding site between the IL-10 and its receptor [6], the identification of T and B cell epitopes [7], and the design of cell-adhesive peptide [8,9], etc.

PEPscan is a particular class of peptide arrays that has been developed to identify, within the sequence of a protein of interest [10], the regions that interact with a known binder. Its principle is to cut a protein or a fragment of a protein, into a series of overlapping peptides of fixed size, and to test each fragment for its binding to the partner protein, peptide, or small molecule. The most common size of peptides for PEPscan ranges from 4 to 20 amino acids, with an overlapping range from 1 to 10 amino acids. One of the first applications of PEPscan was the identification of epitopes recognized by monoclonal antibodies [11]. More recently, PEPscan has gained interest for the modulation of protein-protein interactions, with several reports showing its effectiveness to identify peptides able to interfere with protein-protein interactions [12]. A very recent similar approach concerns the interference between LRRK2 and PP1 in the context of Parkinson’s disease [13]. In these experiments, one hypothesizes that the active peptides identified are located at the protein-protein interface, but in fact very little is known about the exact location of those peptides, relative to the binding interface, as PEPscan is usually employed in cases where the structure of the complex involving the proteins of interest is not known (see [14]). Going farther, one could ask about the utility of PEPscan to assist complex structure modeling. Beyond that, one could also ask if the use of PEPscan could be extended to map protein interactions with other types of partners than proteins, including, for instance, small molecules or peptides.

In this manuscript, we describe the versatility of the PEPscan approach for identification of protein-protein binding sites, we demonstrate that it can effectively be used for the characterization of protein-polysaccharide binding sites, and suggest it can be extended to peptides as well, making it a versatile approach to the characterization of protein-binder interactions.

## 2. Materials and Methods

### 2.1. Protein-Protein Interactions for Complexes of Known Structures

The seven protein complexes of known structure that have been studied are detailed Table 1. They were selected from the Protein-Protein Docking Benchmark 4 [15], with the additional condition that the proteins and antibodies targeting the proteins should be available commercially so as to undergo PEPscan experiments. Complexes involving proteins of small size were favored, and complexes involving antibodies were not considered. The seven complexes in this study display different topologies and involve varied structural classes. All the complexes involve 2 partners, except the EPO/EPOR that involves 3 chains (two being similar), and the SAP/FynSH3 that involves, in addition to the two chains, a small peptide. Of note, for many cases, the 3D structure of the complex does not contain the full protein, but only the domain in interaction. Despite this, the PEPscan experiments were performed using the full-length proteins. For all the complexes, we have performed the PEPscan experiment on both sides, meaning that each partner was split as a series of overlapping fragments of size 12 amino acids on a membrane and hybridized with the other partner of the complex. Note that PEPscan is not required to be performed on both sides, per se. Here, we did this to assess if, and how much, it could be helpful to drive the modeling of protein-protein interactions. Among the peptides identified for one side or the other, it sometimes also occurs that one of the interfering peptides has better biological activity. PEPscan was performed using a standard protocol (see Section 2.2).

### 2.2. Binding Assay on Cellulose-Bound Peptides (PEPscan)

Overlapping dodecapeptides with two amino acid shift, spanning the complete sequence of a protein, were prepared by automatic spot synthesis (Abimed, Langerfeld, Germany) on an amino-derived cellulose membrane, as described. The membrane was saturated using 3% non-fat dry milk/3% BSA (2 h at room temperature), incubated with the purified protein partner (4 mg/mL, 4 °C, overnight), and, after several washing steps, incubated with an antibody against the protein used for the hybridization (2 h at room temperature), followed by an HRP-conjugated secondary antibody for 1 h at room temperature. Positive spots were visualized using the ECL system.

### 2.3. Chondroitin Sulfate Binding Assay on Cellulose-Bound Peptides Containing VAR2CSA Sequence (PEPscan)

Overlapping dodecapeptides with two amino acid shift, spanning a fragment of the VAR2CSA sequence, were prepared by automatic spot synthesis (Abimed, Langerfeld, Germany) on an amino-derived cellulose membrane, as described [1,5]. The membrane was saturated using 3% non-fat dry milk/3% BSA (2 h at room temperature), incubated with chondroitin sulfate (CSA) molecule (5 μg/mL, 4 °C, overnight), and, after several washing steps, incubated with a polyclonal anti-chondroitin sulfate antibody for 2 h at room temperature, followed by an HRP-conjugated secondary antibody for 1 h at room temperature. Positive spots were visualized using the ECL system.

### 2.4. Peptide Synthesis and Sequence

Peptides were synthesized in an automated multiple peptide synthesizer, with solid phase procedure and standard Fmoc chemistry by GL Biochem (Shanghai, China). The purity and composition of the peptides were confirmed by reverse phase high performance liquid chromatography (HPLC) and by mass spectrometry (MS). For some experiments, the peptides were synthesized with the fluorochrome FITC at the N-terminus.

### 2.5. VAR2CSA/CSA Interaction Competition In Vitro

The interaction VAR2CSA/CSA (protein/small molecule) was competed in vitro using 1 mM of chondroitin sulfate A (CSA), incubated for 30 min at room temperature with the peptides identified on the PEPscan approach (peptides 1, 2, and 3). After several washing steps, the CSA/peptide mix was incubated with the membrane containing the VAR2CSA protein, followed by incubation with an anti-CSA antibody, and an HRP-conjugated secondary antibody. The interaction was detected using the ECL system.

### 2.6. FITC-labeled LRRK2 Peptide Binding Assay on Cellulose-Bound Peptides Containing PP1a Sequence (PEPscan)

Overlapping dodecapeptides with two amino acid shift, spanning the complete PP1a sequence, were prepared by automatic spot synthesis (Abimed, Langerfeld, Germany) on an amino-derived cellulose membrane. The membrane was saturated using 3% non-fat dry milk/3% BSA (Sigma-Aldrich, Saint Louis, MI, USA) (2 h at room temperature), incubated with FITC-labeled LRRK2 peptide (10 μg/mL, 4 °C, overnight) and after several washing steps, the membrane was developed using a fluorescence scanner (FITC fluorescence filter).

## 3. Results

### 3.1. PEPscan for Protein-Protein Interactions

We assess here, for seven cases for which the structure of the complex is known, how well PEPscan is able to identify peptides that are located at the protein-protein interface. Figure 1 presents the arrays for both sides for each of the cases, and Table 2 presents the candidate fragments that could be identified from the membranes, and the definition of the Maximally Overlapping Fragments (MOFs), i.e., the fragments that are common to a series of consecutive positive spots (see [14])—for only one spot, we consider no MOF can be identified.

A first observation is a visual heterogeneity in the aspects of the arrays. For some, such as that of TGFβ3 or Ras, there are a large number of positive spots, whereas for others, such as EPOR or NGAL, there are very few. The habit is usually to consider that only a series of at least three consecutive spots can correspond to true positives; we now question this. Looking at Figure 1 and Table 2, one first observes that none of the sequences corresponding to only one spot are located at the PPI. For NCF2/Rac1, only isolated spots and a few series with very little signal are observed. In fact, looking at the structure of NCF2, it seems the interface with Rac1 mostly consists of a few residues, distant in the sequence that belong to loops between the helices. Probably, for interactions involving too few amino acids in a fragment of 12 amino acids, PEPscan may fail to return useful information. Another such case is that of the EPO, again a helical topology, for which only one spot is positive, and does not correspond to the binding interface. For this case however, the spots truly corresponding to the interface are located in a region of the membrane that is hardly interpretable (line 3, spots 23–25), so it is difficult to conclude. Another explanation for this case could be that the experiment itself could fail because the epitope targeted by the antibody is close to or overlaps with the binding interface, and thus cannot bind, preventing spots becoming positive. This was not further investigated. In summary, isolated spots in this study did not lead to useful 3D information. Some series of two consecutive spots do, however, contain information about the binding interface. This is the case for CTLA4, NGAL, and TGFR2, and for all these cases, alternative candidate fragments corresponding to series of at least three consecutive spots are observed, that do not necessarily correspond to the binding interface. Consequently, series of at least two spots should be considered as possible candidates, whereas isolated spots could be discarded safely.

We now turn to membranes with a large number of positive spots, such as Ras/Ral or TGFβ3/TGFR2. For TGFβ3, most of the positive spots are located in a region not resolved in the structure; the 3D structure starting at position 6:2 in the membrane. Three series of consecutive darker spots are observed (6:6–9; 6:15–16; 6:18–20), one (6:6–9) corresponding to the interface. For Ras, only very few spots show no signal. The interaction site corresponds to spots 1:20–23 and does not correspond to the largest series of consecutive spots that are observed on line 2. This suggests that probably, some non-specific binding occurs, and the experimental signal should be filtered using some processing to remove such noise. This is, however, beyond the scope of this study. Although the direct interpretation of the membranes seems to be challenging, we recall that the use of PEPscan for the identification of interfering peptides for instance, requires a confirmation by experiments, such as in vitro competition. We also recall that the reproducibility of PEPscan experiments is fairly good (see [14]).

Overall, in all the cases in this study, fragments associated with MOFs (i.e., corresponding to series of at least two consecutive spots—see Materials and Methods Section) are in limited number among the domains in interaction (30 for 14 proteins), i.e., 2.1 average, ranging from none to eight for NGAL. For this case, four out of the eight fragments identified interact with the partner. Moreover, it seems their number could be reduced to only four, looking at only the most positive (darkest) series (1:13–16, 1:30–31, 2:11–16, 3:5–10).

How well do the fragments overlap the binding site? In terms of the identification of residues at the protein-protein interface, apart from the EPO and NCF2 cases discussed above, it is interesting to note that in all cases, PEPscan could identify fragments involving residues at the interface. Considering the experiments on both sides per complex, we observe that for all cases (100%), PEPscan was able to return information about the binding site for at least one experiment. Considering that only one experiment over the two could be conducted, this rate of success falls to 86% (12/14), which remains high. In total, 46% (18 out of 39) of candidate fragments with MOFs identified in the 3D structures have residues located at the interface. Moreover, as in our previous study, we observe that MOFs that correspond to a subset of these fragments usually contain residues, located at the interface, allowing the number of candidate residues to be narrowed. In total, the ratio of the number of candidate residues of the MOFs at the interface, over that of the MOFs identified in the 3D structures, is 28% (80/282). We recall, however, that PEPscan experiments must be supplemented by in vitro competition experiments to fully identify which fragments effectively interfere with the complex formation.

Finally, we consider the information of the MOFs in terms of the 3D modeling of the complexes, i.e., exploring if the information of the candidate MOFs could help modeling the structures. As can be observed in Figure 1, difficulties arise as the false positive MOFs can correspond to patches very distant from the actual interface. Moreover, even considering only true positives, one observes that the interface residues identified from each of the partners do not necessarily face each other. For instance, looking at the TGFβ3/TGFR2 complex, residues of TGFR2 contacted by the MOFs of TGFβ3 do not correspond to the MOFs of TGFR2, and vice versa. The same observation holds for NGAL/CTLA-4, MMP-1/TIMP1, and Ras/RalGDS. The only case for which the MOFs face each other is that of Fyn SH3/SAP. In summary, if PEPscan can effectively identify fragments at the protein-protein interface, the information obtained from each of the partners is not consistent enough to identify the binding interface in a straightforward manner. This does not preclude that PEPscan experiments could help modeling the structure of the complex, but most probably, it could come from a filtering of series of models, and not from defining restraints to drive the docking. It remains that, in the light of the present results, models for which no MOFs correspond to the interface, should be considered as questionable.

### 3.2. PEPscan for Protein-Polysaccharide Interactions: Identification of VAR2CSA Sequences Involved in Binding to CSA

We now investigate if PEPscan could be extended to the identification of the binding site of small polysaccharides, in conditions similar to that of the protein-protein interface, i.e., an antibody targeting the polysaccharide exists. In order to determine the amino acid residues mediating the binding of the small molecule, CSA, overlapping dodecapeptides covering a fragment of VAR2CSA protein were immobilized in a nitrocellulose membrane and hybridized with CSA (Figure 2A). A set of three darker single spots revealed the presence of three candidate interacting motifs. The three peptides (named 1, 2, and 3) containing the binding sequence of CSA to the protein were chemically synthesized and used for functional analysis.

An in vitro competition assay was performed to confirm that peptides 1, 2, and 3 specifically target the VAR2CSA/CSA interaction. CSA was pre-incubated with 1 mM of each peptide and then the mixture was hybridized with the membrane containing the protein. Figure 2B shows that the interaction of CSA with VAR2CSA is lost upon pre-incubation of CSA with peptides 1 and 2, suggesting that these peptides specifically block the interaction. The sequence corresponding to peptide 3 does not block the interaction, suggesting that this peptide is not specific for VAR2CSA/CSA interaction. In summary, PEPscan identified a small number of three peptide candidates to bind CSA, upon which two seemed to bind CSA while the other did not.

### 3.3. PEPscan for the Identification of the Binding Site of Labeled Peptides

We turn, here, to considering if PEPscan could also be applied to the identification of the binding site of shorter entities, such as peptides, for which no antibody exists, but some means to reveal the positive spots exists, such as labeling. For this, we consider the identification of PP1a sequences involved in binding to one FITC-labeled LRRK2 peptide. We have previously identified and published the sequence of the interfering peptide, blocking the interaction between the kinase LRRK2 and the phosphatase PP1 [13]. As LRRK2 was not available commercially at the time of the study, it was not possible to use the same protocol to identify the peptide of PP1, blocking the interaction with LRRK2. We consider here taking advantage of this sequence (PMGFWSRLINRLLEISPY) to identify the peptide of PP1a, the peptide blocking the interaction of PP1a/LRRK2. The peptide was labeled with FITC and hybridized with a PEPscan membrane containing the sequence of PP1a. The results presented in Figure 3 show a density of plots that is rather similar to that of some membranes in Figure 1, suggesting that some non-specific binding occurs. Nevertheless, it also shows two sets of four and six contiguous PP1a/LRRK2 interacting spots, that could correspond to the binding site of PP1a to LRRK2. Despite remaining to be confirmed, these preliminary results strongly suggest that PEPscan could be extended to the identification of protein-peptide interactions, using labeled peptides. Presently, however, very little is known about the possible impact of the labeling agent.

## 4. Discussion

PEPscan has recently raised new interest in the identification of interfering peptides blocking protein-protein interactions. For partners known to interact, PEPscan is of particular interest when the structure of the partners in interaction is not known, and in silico analysis to design interfering peptides cannot be conducted. The PEPscan approach combines the advantage of being successful, with also being cheap and rapid. We have previously demonstrated PEPscan’s ability to identify peptides at the protein-protein interface [14]. We have re-analyzed the results obtained by PEPscan and compared them to the structural data available and concluded that it identifies candidate fragments on the surface of the protein, in a limited number, and in a rather specific manner. Here, we go further in the analysis of how PEPscan performs in the identification of the protein-protein interface. Considering seven protein-protein interactions for which a structure exists, we show that PEPscan is able to identify candidates located at the protein-protein interface, with a fraction of false positives of only ~0.5, meaning that close to half of the candidate fragments are true positives. Clearly, this makes PEPscan an approach of interest for cases where no structure exists. Moreover, it seems there is room for improvement, particularly from the perspective of using machine learning techniques to reduce the noise when analyzing the membrane. The prospect of using PEPscan to infer constraints to driving the modeling of protein-protein interactions seems, however, more distant. For one thing, recent advances have taken the rate of success of protein-protein complex modeling to an unprecedented level [23]. For another, the motifs identified at the protein-protein interface do not allow the derivation of distance constraints and might provide information that is too ambiguous to be used for anything other than filtering out large ensembles of models. This remains however, a subject for further consideration.

Interestingly, it seems PEPscan can probably be extended to other kinds of interactions involving a protein. For the LRRK2 peptide, we show that a protocol based on fluorescence, directly coupling a small peptide to a marker such as FITC, is likely to be effective—this remains however, to be confirmed. Particularly, the impact of the FITC on peptide conformation and binding is largely unknown. For the CSA, we demonstrate that the protocol can be extended to other categories of binders for which antibodies exist, such as polysaccharides. Indeed, it was possible to confirm the binding of two of the three candidates by in vitro competition. Even if the effectiveness of PEPscan for protein-polysaccharides should be confirmed with more examples, this clearly opens new perspectives for PEPscan.

Overall, the present study confirms that, without any structural knowledge, PEPscan is an approach of interest to characterize the regions of proteins involved in the binding with a partner, and the fragments that could interfere with this binding. This opens the door to the characterization of new interactions involved in pathologies, that fall out of the scope of routine structural biology approaches or structural modeling.

## Figures and Tables

**Figure 1 biomolecules-12-00178-f001:**
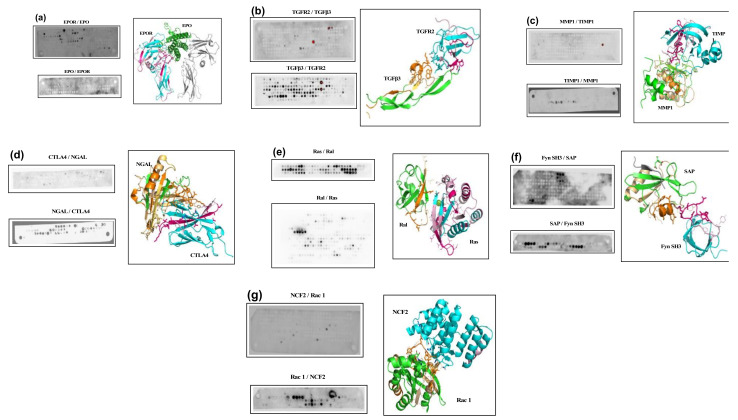
PEPscan applied to protein-protein interactions. Panels A to G correspond to the EPO/EPOR, TGFR2/TGFβ3, MMP1/TIMP1, CTLA4/NGAL, Ras/Ral, Fyn SH3/SAP, and NCF2/Rac 1 complexes, respectively. For each, two PEPscan membranes are presented (X/Y and Y/X), X/Y meaning X corresponds to the membrane and Y to the protein in use for the hybridization. The 3D structure of the complex is depicted, with candidate fragments and MOFs in wheat/orange for partner 1 and purple/pink for partner 2. Side chains of the candidate fragments and the MOFs are depicted as lines and sticks, respectively.

**Figure 2 biomolecules-12-00178-f002:**
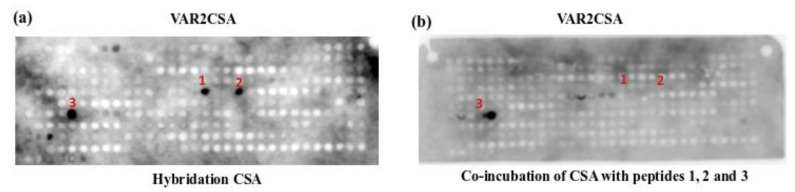
Identification of binding site of CSA to VAR2CSA. (**a**) VAR2CSA membrane hybridized with CSA. (**b**) Competition of the CSA binding to VAR2CSA.

**Figure 3 biomolecules-12-00178-f003:**
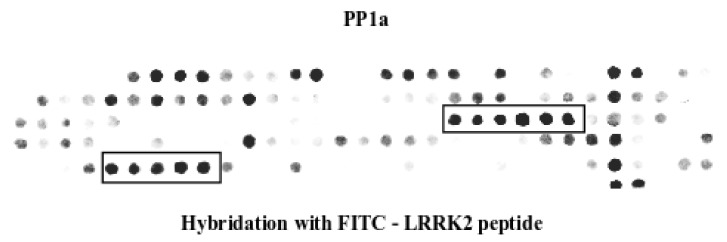
Identification of a binding site of PP1a to FITC-labeled LRRK2 peptide. The sequence of PP1a was developed as series of overlapping dodecapeptides with a shift of two amino acids. The membrane was hybridized with the FITC-labeled LRRK2 peptide that interacts with PP1a. Spots were detected by fluorescence. The peptides interacting with PP1a are boxed and the sequence shown.

**Table 1 biomolecules-12-00178-t001:** Protein-protein complexes of a known structure considered in this study. For each, we detail the PDB identifier and the associated publication, the chains corresponding to the partners, their Uniprot identifiers, and the size of the sequences of the protein as in Uniprot and as in the structure. TGFβ: Tumor Growth Factor beta, TGFR: Tumor Growth Factor Receptor, EPO: Erythropoietin, EPOR: Erythropoietin Receptor, NCF2: Neutrophil Cytosolic Factor, NGAL: Neutrophil Gelatinase-Associated Lipocalin, SAP: SLAM-Associated Protein, MMP1: Matrix Metalloproteinase-1 Polymorphism.

Complex	PDB Id.	PDB Chain	PDB Size	Protein	Uniprot Id.	Full Size
NCF2/Rac1	1e96 [16]	A	178	Rac1	P63000	192
		B	185	NCF2	P19878	526
SAP/FynSH3	1m27 [17]	A	105	SAP	O60880	128
		C	61	FynSH3	P06241	537
RalGDS/Ras	1lfd [18]	A	87	RalGDS	Q03386	895
		B	167	Ras	P01112	189
MMP-1/TIMP1	2j0t [19]	A	161	MMP-1	P03956	469
		D	124	TIMP-1	P01033	207
TGFbetaR2/TGF-beta3	1ktz [20]	B	106	TGFbetaR2	P37173	567
		A	82	TGF-beta3	P10600	412
NGAL/CTLA-4	3bx7 [21]	A	173	NGAL	P80188	198
		C	120	CTLA-4	P16410	223
EPO/EPOR	1eer [22]	A	166	EPO	P01588	193
		B	213	EPOR	P19235	508

**Table 2 biomolecules-12-00178-t002:** Candidate fragments identified by PEPscan. For each protein, we report the positions that can be considered as positive in the membrane (x:y–z meaning line x, columns y to z), and the corresponding amino acid sequence, in which MOFs are in **bold**. Fragments at the 3D interface are labeled using a *. Protein fragments for which no 3D coordinates are available are labeled as “**no 3D**”.

PPI	Fragments		PPI	Fragments	
NCF2	3:14	KSEPRHSKIDKA	TGFR2	1:26–27 2:20–22 3:13–14 4:26–27 5:19 6:5–7 6:20–23 7:15–17	CK**FCDVRFSTCD**NQ * CHDP**KLPYHDFI**LEDA FS**EEYNTSNPDL**LL HN**TELLPIELDT**LV (no 3D) SWKTEKDIFSD (no 3D) KQYW**LITAFHAK**GNLQ (no 3D) KLGSSL**ARGIAH**LHSDHT (no 3D) SLRL**DPTLSVDD**LANS (no 3D)
	4:13–14	QDSFSGFAPLQPQAAE (no 3D)			
	5:23	YLEPVELRIHPQ (no 3D)			
	6:7–9	SKAPGRPQLSPGQKQK (no 3D)			
	7:12	RPRDSNELVPLS (no 3D)			
	8:20	PEDLEFQEGDII (no 3D)			
	9:8	VEDCATTDLEST (no 3D)			
RAC1	1:11–14	ISYTTNAFPGEYIPTVFD *	TGFβ3	1:12–14 1:23–25 3:3–6 3:24–25 4:1–3 4:10–14 4:20–22 5:12–14 5:16–19 6:6–9 6:15–16 6:18–20	SLST**CTTLDFGH**IKKK (no 3D) GQIL**SKLRLTSP**PEPT (no 3D) SKVF**RFNVSSVE**KNRT (no 3D) FQ**ILRPDEHIAK**QR (no 3D) GGKN**LPTRGTAE**WLSF (no 3D) TDTVREWL**LRRE**SNLGLEIS (no 3D) IHCP**CHTFQPNG**DILE (no 3D) KDHH**NPHLILMM**IPPH (no 3D) ILMMIP**PHRLDNPG**QGRK (no 3D) YIDF**RQDLGWKW**VHEPKG * YY**ANFCSGPCPY**LR SGPC**PYLRSADT**THST
	1:26–29	VNLGLWDTAGQEDYDR			
	2:17–19	AKWYPEVRHHCPNTPI			
	3:15–18	KYLECSALTQRGLKTVFD *			
SAP	1:3–6	AVYHGKISRETGEKLLLA	NGAL	1:13–16 1:30–31 2:2–6 2:11–16 2:23:27 3:5–10 3:13–17 3:22–24	SDLIPA**PPLSKV**PLQQNF NA**ILREDKDPQK**MY * DKDPQKMY**ATIY**ELKEDKSY SYNVTSVLFR**KK**KCDYWIRTFV * CQPGEFTL**GNIK**SYPGLTSY TNYNQHAMVF**FK**KVSQNREYFK * NREYFKIT**LYGR**TKELTSEL * ELKE**NFIRFSKS**LGLP
	1:13–15	LDGSYLLRDSESVPGV			
	2:8–10	FRKIKNLISAFQKPDQ *			
	2:20–23	PVEKKSSARSTQGTTGIR			
FSYN-SH3	1:14–16	GYRYGTDPTPQHYPSF (no 3D)	CTLA4	1:30–31 2:3–5	EY**ASPGKATEVR**VT * TEVR**VTVLRQAD**SQVT *
	2:14–16	ALYDYEARTEDDLSFH *			
	3:14–16	FGKLGRKDAERQLLSF (no 3D)			
	9:3–5	HCWKKDPEERPTFEYL (no 3D)			
RAS	1:6–9	AGGVGKSALTIQLIQN	EPO	1:11	LGLPVLGAPPRL (no full 3D)
	1:20–23	SYRKQVVIDGETCLLD *			
	2:1–8	EYSAMRDQYMRTGEGFLCVFAINNTK			
	2:15–18	EDIHQYREQIKRVKDSDD			
	2:23–28	DDVPMVLVGNKCDLAARTVESR			
	3:6–9	RSYGIPYIETSAKTRQGV			
	3:24–28	LNPPDESGPGCMSCKCVLLS (no 3D)			
RAL	6:3–7	PTLAPAPELDPTVSQSLHLE (no 3D)	EPOR	1:10 1:16 2:1 2:14–16 3:6–8 4:8–9 4:17–18 5:13–16	LAGAAWAPPPNL (no 3D) PDPKFESKAALL FWEEAASAGVGP KLCR**LHQAPTAR**GAVR RVIH**INEVVLLD**APVG * GR**TECVLSNLRG**RT AV**RARMAEPSFG**GF * SHRRAL**KQKIWP**GIPSPE (no 3D)
	13:18–20	DCCIIRVSLDVDNGNM *			
TIMP1	2:6–11	QALGDAADIRFVYTPAMESVCG * MESV**CGYFHRSHNR**SEEF			
	2:14–17				
MMP1	3:3	DVDHAIEKAFQL			
	4:1	PGPGIGGDAHFD			
	4:11–14	FREYNLHRVAAHELGHSL *			
	4:28	LMYPSYTFSGDV			
	7:19–20	PGYPKMIAHDFPGI (no 3D)			

## Data Availability

Not applicable.

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
