# Peer review of "PEPscan: A Broad Spectrum Approach for the Characterization of Protein-Binder Interactions?"

_biomolecules, 2022, doi:10.3390/biom12020178_

Round 1

Reviewer 1 Report

How to screen the ligands of protein interaction and how to analyze and identify the binding between two proteins or between proteins and other molecules has always been the focus of researchers in this field. In this research paper, PEP scan is used to analyze the binding of protein and polypeptide, protein and chondroitin sulfate, and give a good attempt to analyze the competitive binding polypeptide between them. It has a good reference for promoting the study of protein-protein interaction. Here are some of our comments on this article.

  1. The 7 protein complexes mentioned in the article should have corresponding literature references, and some abbreviations in the article should be full names.
  2. What does the full name of "CS" mean
  3. Line216 "()" should be "(b)"
  4. Line247 (Ref PLoS One) should be specific to references
  5. Where is the "standard protocol" of line76?

Author Response

How to screen the ligands of protein interaction and how to analyze and identify the binding between two proteins or between proteins and other molecules has always been the focus of researchers in this field. In this research paper, PEP scan is used to analyze the binding of protein and polypeptide, protein and chondroitin sulfate, and give a good attempt to analyze the competitive binding polypeptide between them. It has a good reference for promoting the study of protein-protein interaction. Here are some of our comments on this article.

The 7 protein complexes mentioned in the article should have corresponding literature references, and some abbreviations in the article should be full names.
Answer: We have added the references for the seven complexes. When possible the full names of the proteins have been detailed in Table 1 caption.

What does the full name of "CS" mean
Answer: We thank the referee for pointing to this error. CS was an incorrect abbreviation for CSA. 

Line216 "()" should be "(b)"
Answer: We have fixed the typo.

Line247 (Ref PLoS One) should be specific to references
Answer: We apologize for this omission. Reference [13] is now cited instead.

Where is the "standard protocol" of line76?
Answer: PEPscan is defined in section 2.2: "Binding assay on cellulose-bound peptides (PEPscan)".  We have modified the text to prevent any confusion: "(see section 2.2).

Reviewer 2 Report

The manuscript by Rebollo et al describes a method “PEPscan” as an approach for identifying new PPI. However, the manuscript benefits from improvement in some aspects before it can be published.

MAJOR

1) How much does the method help finding new PPI?

The main idea of the Pepscan method can be stated as:

PEPscan can provide a cheap and rapid 8 manner to identify candidate interfering peptides (IPs), i.e., peptides able to disrupt a target protein-protein interaction.

PEPscan has recently raised new interest for the identification of interfering peptides blocking protein/protein interactions. It is interesting when the structure of the partners interaction is not known

(both are quotes from the current manuscript).

Considering the current manuscript and the previous publication, the most crucial question is this: if and how much does the method help finding new PPI? The current manuscript does not help too much answering this question. In both the manuscript and the publication, only known PPI partners are used as validation (simulation) of the method. However, the reader should have more clear understanding of the potential use of the method and what should be expected when potentially using the method for unknown PPI partners.

In the current manuscript (and previous publication), known PPI partners are used and only known partners are hybridized against each other. What is the expected success rate and which success can be reasonably expected at all? Is it feasible for unknown PPI partners? Can we only identify potential partners or can we also estimate specific interaction surfaces (i.e. areas of the primary sequence)? Do we need to do the experiment in a 2-way (both partners are fragmented), as was conducted here? Does 2-way increase the probability of true pos success and how much? Can we expect a certain success % of confirming an interaction between 2 unknown protein partners? Or could we predict (with reasonable probability) the success of identifying specific binding fragments? (ref: “PEPscan could identify fragments involving residues at the interface. 46%” and ref: “In summary, if PEPscan can effectively 201 identify fragments at protein-protein interface the information obtained from each of the 202 partner is not consistent enough to identify the binding interface in a straight manner”).

Discussing, opening, and elaborating these questions would give the reader information that allow the use of the method and decide if this method should be used or not.

2) How were the analyzed PPI chosen?

As was said in the manuscript, the chosen partners had known data about the PPI and 3d structure. Theoretically, the analysis of the current manuscript can be affected by the choice of the input protein partners (i.e. inclusion or exclusion of successful or unsuccessful experiments with Pepscan). In the current manuscript, as there are PPI experiments included that can be considered “unsuccessful” (the ones that did not yield many positive spots), there of course is no doubt that the selection was unbiased, but the definition or explanation of HOW the initial selection of PPI was made should be still added and clarified.

Minor

Figure 1 has a bad quality, it can not be zoomed in to take a closer look. Looking at it without zooming one can not identify the positive spots clearly.

Line 266, line 247 reference format

Typos at lines 295, 65, 66

Line 83 ref missing.

Author Response

The manuscript by Rebollo et al describes a method “PEPscan” as an approach for identifying new PPI. However, the manuscript benefits from improvement in some aspects before it can be published.

MAJOR

1) How much does the method help finding new PPI?
The main idea of the Pepscan method can be stated as:

PEPscan can provide a cheap and rapid 8 manner to identify candidate interfering peptides (IPs), i.e., peptides able to disrupt a target protein-protein interaction.

PEPscan has recently raised new interest for the identification of interfering peptides blocking protein/protein interactions. It is interesting when the structure of the partners interaction is not known

(both are quotes from the current manuscript).

Considering the current manuscript and the previous publication, the most crucial question is this: if and how much does the method help finding new PPI? The current manuscript does not help too much answering this question. In both the manuscript and the publication, only known PPI partners are used as validation (simulation) of the method. However, the reader should have more clear understanding of the potential use of the method and what should be expected when potentially using the method for unknown PPI partners.

Answer: We are unsure we understand this remark. We have never claimed that PSPscan helps discovering new PPIs, because it is not its goal. As stated in the manuscript, it can help to identify interfering peptides, i.e. peptide competing with the formation of a "target" complex, i.e. given the interaction has been demonstrated previously. The advantage of the PEPscan is here that it returns information even if no information about the way the proteins interact is known. This was stated in the introduction. To make it even clearer we now state in the introduction: "PEPscan is a particular class of peptide arrays that has been developed to identify, within the sequence of a protein of interest [10], the regions that interact with a known binder.", and in the Discussion: "

1) How much does the method help finding new PPI?
The main idea of the Pepscan method can be stated as:

PEPscan can provide a cheap and rapid 8 manner to identify candidate interfering peptides (IPs), i.e., peptides able to disrupt a target protein-protein interaction.

PEPscan has recently raised new interest for the identification of interfering peptides blocking protein/protein interactions. It is interesting when the structure of the partners interaction is not known

(both are quotes from the current manuscript).

Considering the current manuscript and the previous publication, the most crucial question is this: if and how much does the method help finding new PPI? The current manuscript does not help too much answering this question. In both the manuscript and the publication, only known PPI partners are used as validation (simulation) of the method. However, the reader should have more clear understanding of the potential use of the method and what should be expected when potentially using the method for unknown PPI partners.

Answer: We are unsure we understand this remark. We have never claimed that PSPscan helps discovering new PPIs, because it is not its goal. As stated in the manuscript, it can help to identify interfering peptides, i.e. peptide competing with the formation of a "target" complex, i.e. given the interaction has been demonstrated previously. The advantage of the PEPscan is here that it returns information even if no information about the way the proteins interact is known. This was stated in the introduction. To make it even clearer we now state in the introduction: "PEPscan is a particular class of peptide arrays that has been developed to identify, within the sequence of a protein of interest [10], the regions that interact with a known binder.", and in the Discussion: "For partners known to interact, PEPscan is of particular interest when the structure of the partners interaction is not known and in silico analysis to design interfering peptides cannot be undergone."

In the current manuscript (and previous publication), known PPI partners are used and only known partners are hybridized against each other. What is the expected success rate and which success can be reasonably expected at all? Is it feasible for unknown PPI partners? Can we only identify potential partners or can we also estimate specific interaction surfaces (i.e. areas of the primary sequence)? Do we need to do the experiment in a 2-way (both partners are fragmented), as was conducted here? Does 2-way increase the probability of true pos success and how much? Can we expect a certain success % of confirming an interaction between 2 unknown protein partners? Or could we predict (with reasonable probability) the success of identifying specific binding fragments? (ref: “PEPscan could identify fragments involving residues at the interface. 46%” and ref: “In summary, if PEPscan can effectively 201 identify fragments at protein-protein interface the information obtained from each of the 202 partner is not consistent enough to identify the binding interface in a straight manner”).

Discussing, opening, and elaborating these questions would give the reader information that allow the use of the method and decide if this method should be used or not.

Answer: Again, PEPscan is not feasible for unknown PPI partners (see our previous answers). Considering known partners, we perform here experiments both sides to assess if PEPscan could help modeling proteins in interaction. We now recall this in section 2.1: "Note that PEPscan does not requires per se, to be performed both sides. Here, we did this to assess if and how much it could be helpful to drive the modeling of protein-protein interactions. Among the peptides identified for one side or the other, it also occurs sometimes that one of the interfering peptides has better biological activity.". Our conclusions are that the indications provided by PEPscan seem too fuzzy to drive protein-protein docking up to high quality models. Instead, it could probably be used to filter out models, but this is not the focus of the present contribution.
Considering the rate of success of PEPscan, we already had a paragraph detailling this in section 3.1:
" How well do the fragments overlap the binding site? In terms of the identification of residues at the protein-protein interface, apart from the EPO and NCF2 cases discussed above, it is interesting to note that in all cases, PEPscan could identify fragments involving residues at the interface. 46% (18 out of 39) candidate fragments with MOFs identified in the 3D structures have residues located at the interface. Moreover, as in our previous study, we observe that MOFs that correspond to a subset of these fragments usually contain the residues located at the interface, allowing to narrow the number of candidate residues. In total the ratio of the number of candidate residues of the MOFs at the interface over that of the MOFs identified in the 3D structures is of  28% (80/282)."
We understand from the referee's comments that some more information is requested, particularly at the level of experiments, and considering or not both side experiments. We now state in addition:
" Considering both side experiments per complex, we observe that for all cases (100%), PEPscan was able to return information about the binding site for at least one experiment. Considering that only one experiment over the two could be done, this rate of success falls down to 86% (12/14), which remains high". and "We recall however that PEPscan experiments must be supplemented by in vitro competition experiments to fully identify which fragments effectively interfere with the complex formation.". 

2) How were the analyzed PPI chosen?
As was said in the manuscript, the chosen partners had known data about the PPI and 3d structure. Theoretically, the analysis of the current manuscript can be affected by the choice of the input protein partners (i.e. inclusion or exclusion of successful or unsuccessful experiments with Pepscan). In the current manuscript, as there are PPI experiments included that can be considered “unsuccessful” (the ones that did not yield many positive spots), there of course is no doubt that the selection was unbiased, but the definition or explanation of HOW the initial selection of PPI was made should be still added and clarified.

Answer: How the complexes were chosen was already detailed in section 2.1, particularly the need to have the proteins and antibodies targeting them available commercially. We have extended a bit this section: "Complexes involving proteins of small size were favored, and complexes involving antibodies were not considered. The seven complexes of this study display different topologies and involve varied structural classes."

Minor
Figure 1 has a bad quality, it can not be zoomed in to take a closer look. Looking at it without zooming one can not identify the positive spots clearly.
Answer: We agree with the referee. For this reason, we had made full high quality images available separately during the submission process. For the revised version, we have changed the image format in the manuscript which we hope, improves its readability. 

Line 266, line 247 reference format
Typos at lines 295, 65, 66
Answer: We apologize for these mistakes. They have been fixed. 

Line 83 ref missing.
Answer: We are unsure if we understand correctly. Line 83, it is the name of an equipment that is given, not a reference.